# *Trichosporon asahii*: A Potential Growth Promoter for *C. gallinacea*? Implications for Chlamydial Infections and Cell Culture

**DOI:** 10.3390/microorganisms13020288

**Published:** 2025-01-27

**Authors:** Erika Ornelas-Eusebio, Fabien Vorimore, Rachid Aaziz, Maria-Lucia Mandola, Francesca Rizzo, Monica Marchino, Chiara Nogarol, Veronica Risco-Castillo, Gina Zanella, Christiane Schnee, Konrad Sachse, Karine Laroucau

**Affiliations:** 1University Paris-Est, French Agency for Food, Environmental and Occupational Health & Safety (Anses), Animal Health Laboratory, Bacterial Zoonoses Unit, 94700 Maisons-Alfort, France; erika.o.e@gmail.com (E.O.-E.); fabien.vorimore@anses.fr (F.V.); rachid.aaziz@anses.fr (R.A.); 2Istituto Zooprofilattico Sperimentale del Piemonte, Liguria e Valle d’Aosta, Specialistic Virology Unit, 10154 Torino, Italy; marialucia.mandola@izsplv.it (M.-L.M.); chiara.nogarol@izsplv.it (C.N.); 3Jockey Club College of Veterinary Medicine and Life Sciences, City University of Hong Kong, Kowloon, Hong Kong; fragiol@libero.it; 4Servizio Veterinario ASL TO5, S.C. Sanità Animale, 10023 Chieri, Italy; marchino.monica@aslto5.piemonte.it; 5Anses, INRAE, Ecole Nationale Vétérinaire d’Alfort, UMR BIPAR, Laboratoire de Santé Animale, 94700 Maisons-Alfort, France; veronica.risco-castillo@vet-alfort.fr; 6University Paris-Est, Anses, Animal Health Laboratory, Epidemiological Unit, 94700 Maisons-Alfort, France; gina.zanella@anses.fr; 7Friedrich-Loeffler-Institut (FLI) (Federal Research Institute for Animal Health), Institute of Molecular Pathogenesis, 07743 Jena, Germany; christiane.schnee@fli.de; 8RNA Bioinformatics and High-Throughput Analysis, Friedrich-Schiller-Universität Jena, 07743 Jena, Germany; konrad.sachse@gmx.net

**Keywords:** *Chlamydia gallinacea*, cultivation, optimization, *Trichosporon*

## Abstract

The cultivation of *Chlamydia gallinacea*, a recently identified species, is challenging due to the lack of an optimized protocol. In this study, several infection protocols were tested, including different cell lines, incubation temperatures, centrifugation methods and culture media. However, none were successful in field samples. The only exception was a chance co-culture with *Trichosporon asahii*, a microorganism commonly found in the chicken gut. This suggests that current in vitro methods may not be optimized for this species and that host-associated microorganisms may influence the in vivo growth of *C. gallinacea*, which is typically found in the chicken gut. These findings raise new questions and highlight the need for further investigation of microbial interactions within the host, particularly to understand their role in the proliferation of chlamydial species.

## Short Communication

*Chlamydiae* are Gram-negative bacteria that are obligate intracellular parasites with a unique life cycle in which extracellular infectious elementary bodies facilitate dissemination while intracellular non-infectious reticulate bodies are responsible for replication. The family *Chlamydiaceae* comprises two genera (*Chlamydia* and *Chlamydiifrater*) and 11 species within the genus *Chlamydia*, some of which cause infections in humans and animals [1]. *C. gallinacea* is widely distributed in poultry [2] and has recently been detected in poultry farmers, suggesting a potential risk of bird-to-human transmission [3].

In vitro growth of *chlamydiae* depends on suitable conditions for the attachment of elementary bodies to host cell membranes and a favourable intracellular environment for bacterial development. Various methods have been used to optimize chlamydial growth protocols, such as the chemical or physical treatment of cell cultures, incubation temperature optimization, and specific cell lines [4,5]. However, some chlamydial strains and species remain difficult to cultivate, such as *C. suis* from pigs [6,7], *C. pecorum* from the ruminant intestinal tract, *C. psittaci* from non-avian sources [8], and *C. pneumoniae* from human respiratory specimens [9].

The first isolates of *C. gallinacea* were obtained from cloacal swabs of chickens in France using chicken embryos as a culture method [10]. One of these isolates (08-1274/3) was propagated in the BGM cell line to study its ultrastructural and molecular characteristics [11]. However, in subsequent studies in different laboratories, the infectivity tended to decrease with each passage in chicken embryos or BGM cells, with small diffuse inclusions observed by immunofluorescence in some laboratories, leading to strain loss (unpublished data).

The aim of this study was to test different culture conditions for isolating *C. gallinacea* from field samples, using immunofluorescence to visualize well-formed inclusions. This method reliably detects chlamydial inclusions in cultured cells, allowing for the assessment of strain viability and further detailed strain characterization.

A standard protocol involving immunofluorescence labelling is commonly used for the isolation of *Chlamydiaceae* [12]. Briefly, after inoculation, the BGM cell monolayers were centrifuged at 3000× *g* for 1 h at 37 °C, followed by 2 h at 37 °C in 5% CO_2_. The cell growth medium was replaced with serum-free UltraMDCK supplemented with 1% non-essential amino acids, 1% vitamins and antibiotics/antifungals (25 µg/mL of vancomycin, 10 µg/mL of gentamycin, 2.5 µg/mL of amphotericin B and 25 U/mL of nystatin). This medium was changed after 24 h, and the infected cells were then incubated at 37 °C without CO_2_. Inclusions are usually visualized 48 h post-infection (pi) by immunostaining using the IMAGEN Chlamydia kit (Oxoid Ltd., Cambridgeshire, UK). Up to two passages were performed by collecting an infected vial at 48 h pi, freezing at −80 °C, sonication (sonication amplitude 80%, 0.8 s 10 cycles and 0.2 s pause between each cycle), and reinoculating 100 µL into a new vial containing fresh BGM cells.

We applied this protocol to four *C. gallinacea*-positive cloacal samples from Mexican poultry farms (A1, A8, Pue6-6 and Pue6-7, Table 1) [13]. However, cultivation resulted in poorly formed inclusion-like structures that were lost after subsequent passages (Figure 1).

To optimize the culture of *C. gallinacea*, we made specific modifications to the standard protocol and tested them on the *C. gallinacea* reference strain 08-1274/3_08DC63, which was initially grown on chicken embryos at Anses (Maisons-Alfort) and successfully propagated on cell lines at FLI (Jena). An overview of the optimization experiments is shown in Figure A1. The modified parameters included the cell lines (Experiment 1), the incubation temperature (Experiment 2), and the addition of a daily centrifugation step (Experiment 3).

Experiment 1: BGM (ATCC PTA-4594), DF-1 (ATCC CRL-3586), and Caco-2 (ATCC HTB-37) cell lines were compared. DF-1, derived from chicken embryo fibroblasts, shows higher mitochondrial function and faster division than other fibroblasts. Caco-2, derived from a colon adenocarcinoma, is intestinal. The aim of this comparison was to identify the best cell line for *C. gallinacea*, given its intestinal tropism in birds. The culture conditions for these three cell lines are described in Table A1. The susceptibility of these cell lines to infection was assessed using the titrated *C. gallinacea* strain 08-1274/3_08DC63 (7.3 × 10^7^ IFU/mL). Well-defined chlamydial inclusions were observed in infected cells at 48 h pi with all cell lines. After passage, the size of the chlamydial inclusions varied between the cell lines, with Caco-2 cells showing larger inclusions than the other two cell lines. BGM cells showed chlamydial inclusions that were still visible after several days of infection in a nearly intact cell monolayer, whereas cell detachment was observed in DF-1 and Caco-2 cultures, probably due to a fast cell growth rate and the production of overlapping layers compared to BGM cells (Figure A2). Unfortunately, these cells detached during immunostaining, resulting in holes in the cultures and the loss of the immunofluorescence signal. Therefore, we decided to conserve the BGM cells in our experimental protocol.

Experiment 2: The effect of incubation temperature on inclusion formation was investigated because some chlamydial species, such as *C. poikilothermis*, have been shown to grow better at temperatures similar to those found in their natural hosts [17]. As birds have a natural temperature range between 38 °C and 43 °C, and to avoid exceeding the temperature limit set for BGM cells, we tested the *C. gallinacea* culture derived from Experiment 1 at 37 °C and 39 °C. When comparing these two incubation temperatures, we did not observe any difference in the growth of *C. gallinacea*, so we decided to maintain a temperature of 37 °C in our experimental protocol, as this is the classical temperature used for the cultivation of the *Chlamydiaceae* on this cell line.

Experiment 3: The addition of a daily centrifugation step instead of a single centrifugation was tested in our protocol, as this has been shown to improve the culture of *C. pneumoniae* [18], by promoting the chlamydial attachment to the cells. For this, BGM cells were infected with the *C. gallinacea* culture derived from Experiment 1 and incubated at 37 °C for up to 7 days. Countless inclusion-like, irregularly sized, bright green chlamydial structures were observed in the cytoplasm of infected cells, with no visible difference in *C. gallinacea* growth between the two protocols used (up to 168 h pi) (Figure A3). As this experiment showed no negative effect of daily centrifugation on *C. gallinacea* cultures, we decided to include this step in our protocol without renewing the culture medium, as it may have facilitated the entry of the *Chlamydiaceae* into the cells.

The comparison of different parameters (Experiments 1–3) for the growth of the *C. gallinacea* strain 08-1274/3_08DC63 showed no significant changes in inclusion formation compared to the standard protocol. We therefore decided to maintain the BGM cells at 37 °C with daily centrifugation. As a final approach, we tested an enriched culture medium inspired by Donati and colleagues [4] for direct application to field samples. After inoculation and centrifugation, the cell culture medium was replaced with EMEM supplemented with glucose, L-glutamine, cycloheximide and 20% of fetal calf serum (Table A2). This alternative protocol required CO_2_ incubation after medium replacement.

As the four positive Mexican cloacal swabs were not available for further cultivation attempts, the alternative protocol was applied to four freshly collected *C. gallinacea*-positive samples obtained from cloacal swabs of chickens from Italy [3]. These samples were collected in duplicate, with one sample tested by PCR and the second stored in preservation buffer at −80 °C until processing (Table 1). After their inoculation, immunostaining performed at 48 h pi showed innumerable inclusion-like, irregularly sized, bright green chlamydial structures in the cytoplasm of cells infected with all Italian samples, comparable to the results for Mexican samples using the standard protocol. Immunostaining of the first passages showed minimal differences in *C. gallinacea* growth in three of the four samples (41645-5, 44638-9, and 67320-6) (Figure 1), as seen in the Mexican samples using the standard protocol. However, one sample (60260-3) showed atypical development of chlamydial growth, with a few medium-sized, well-defined, light-green chlamydial inclusions observed at 48 h pi during the first passage. All samples were subcultured according to the normal procedure, and the chlamydial growth of all but one sample showed a pattern similar to that of the Mexican samples. In fact, only in sample 60260-3 were there several large, well-defined and light-green typical chlamydial inclusions observed, occupying almost the entire cell cytoplasm at 48 h pi of the second passage (Figure 1). The presence of *C. gallinacea* in sample 60260-3 was confirmed by real-time PCR [15]. Interestingly, the culture was found to be co-infected with a fungus, with macroscopically observable hyphal-like structures present within the first 24 h pi (Figure 2). Despite the renewal of the chlamydial infection medium (with antibiotics and amphotericin B) every 24 h and the passages, the fungal contamination persisted even after the addition of a fourfold concentration of antifungals to the culture medium. The fungus was grown on Sabouraud agar (Figure 2) and then identified as the dimorphic yeast *Trichosporon asahii* by matrix-assisted laser desorption/ionization time-of-flight mass spectrometry.

*Trichosporon* species are common in nature, with isolates obtained from various sources such as soil, wood, water bodies, animals, and birds [19]. In chickens, *Microascus* sp., *Trichosporon* spp., and *Aspergillus* spp. account for more than 80% of the total fungal population diversity, with *T. asahii* being the predominant species within the genus *Trichosporon* [20]. The detection of *Trichosporon* spp. in the culture of an atypically growing *C. gallinacea* isolate raises questions about its effect on the growth of this chlamydial strain. Co-infections of certain microorganisms with *Chlamydia* have been shown to have synergistic effects that may be beneficial to one or both pathogens, or prevent the growth of one pathogen by another. Previous studies have reported natural chlamydial co-infections with other viral and bacterial pathogens in birds [21,22,23]. A recent study has shown that the biofilm formed by *Candida albicans* may favour the survival of *C. trachomatis* within it, thereby maintaining its infectious properties [24]. This biofilm provides a protective environment that reduces chlamydial susceptibility to antibiotics and promotes evasion of the host immune system. Several *Trichosporon* species, including *Trichosporon asahii*, are capable of producing highly resistant biofilms [25]. In humans, co-infections with *Trichosporon* have been reported, mainly in immunocompromised patients, in association with other viruses or bacteria [26,27,28].

*Trichosporon asahii* and its metabolites could also potentially interfere with chlamydial growth at several levels, such as activating chlamydial genes, providing nutrients necessary for chlamydial growth (e.g., energy molecules and/or co-factors), or facilitating chlamydial penetration into the cell. Indeed, *Trichosporon* is known to produce and secrete enzymes such as proteases and phospholipases that scavenge nutrients from the environment or facilitate the invasion process into various host tissues [19]. *Trichosporon* also expresses the immunomodulatory polysaccharide glucuronoxylomannan (GXM) in its cell wall, which is composed of mannose, xylose, glucose, and glucuronic acid [29], and this GXM could serve as an energy source for *Chlamydia*.

The in vitro observation that *C. gallinacea* strains are lost after repeated passages may be due to the depletion of essential metabolites or compounds necessary for *C. gallinacea* infectivity. This could also explain why inclusions are observed during the initial cultivation of a field sample but disappear during subsequent passages. The *C. gallinacea* reference strain has been successfully propagated on a BGM cell line for up to 10 passages at the FLI in Germany, in contrast to the results obtained at Anses in France using an identical protocol (standard protocol). This raises the possibility that small variations in sensitive reagents, such as fetal calf serum (which may differ in protein, vitamin, or mineral content), could be critical for *C. gallinacea* growth. These essential nutrients could be provided in vivo by *Trichosporon*, a yeast naturally present in poultry, in the context of co-culture, as identified in our study. While yeast extracts are rarely used to culture *Chlamydia*, they are commonly used for mycoplasmas, which also have small genomes and limited biosynthetic capabilities. Growth factors provided by yeast extracts or fetal calf sera [30] are not standardized and are likely to have varying nutrient content between batches, potentially leading to inconsistent results. Transcriptomic studies could help to identify the necessary supplements to be added, as recently described for *Mycoplasma ovipneumoniae* [31].

In conclusion, the experimental protocols used to optimize the growth of *C. gallinacea* did not lead to improved cultivation methods for better inclusion formation and passage propagation. However, despite laboratory constraints that prevented an evaluation of the alternative protocol with the reference strain of *C. gallinacea*— a limitation of this study—successful propagation of a strain was achieved in a field sample. This sample showed well-formed inclusions that multiplied through passages, consistent with observations in other chlamydial species. The differences in growth patterns cannot be attributed to the initial bacterial load, as their Cq values were similar, or to storage conditions. It is likely that the co-culture of a naturally occurring amphotericin B-resistant yeast in the field sample, which was not eliminated by the antibiotic/antifungal cocktail in the culture medium, influenced the growth of *C. gallinacea* through passages. Our results suggest that co-culture with a specific fungus may provide favourable conditions for the growth of *C. gallinacea* in cell culture, as the presence of the fungus may have provided the necessary substrates or conditions for *C. gallinacea* growth in vitro.

It is now important to investigate how the yeast-depleted strain behaves in vitro and to assess its effect on other *C. gallinacea* strains. In addition, it would be valuable to determine whether this effect is specific to *Trichosporon* or whether other yeasts could have a similar effect. Several *Chlamydia* species with gastrointestinal tropism have recently been described in birds, suggesting host specificity (e.g., C. *gallinacea* in chickens, *Candidatus* C. ibidis in flamingos, and *C. buteonis* in birds of prey). Given their small genomes, *Chlamydia* species may have evolved to adapt to their environment, in particular to their host’s gut microbiota and available resources. This discovery provides insights into how Chlamydia species may have evolved in relation to their hosts, highlighting the potential role of host microbiota and environmental factors in shaping their specificity and growth strategies.

From a veterinary and public health perspective, it would be important to assess whether and how fungal co-infections in vivo might affect the infectivity of *Chlamydia* species in their hosts. In particular, a recent case of pneumonia caused by *C. psittaci* and *Cryptococcus* infection was identified by WGS in a farmer who had adopted a stray pigeon prior to the onset of symptoms [32]. Further research is needed to clarify this possible synergistic interaction.

## Figures and Tables

**Figure 1 microorganisms-13-00288-f001:**
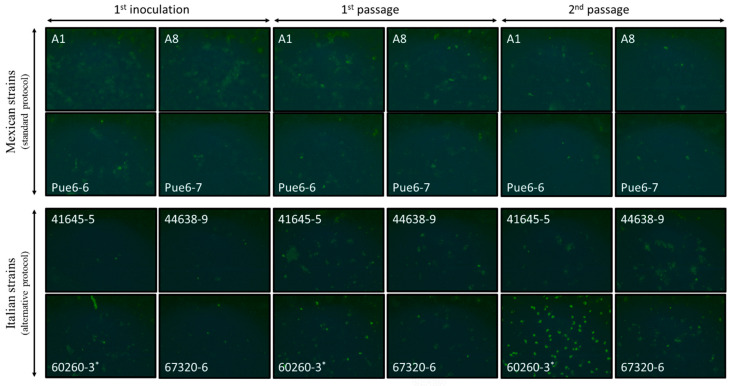
Photomicrographs of the BGM cells inoculated with 4 *C. gallinacea* field samples from Mexico (samples A1, A8, Pue6-6 and Pue6-7) and 4 *C. gallinacea* field samples from Italy (samples 41645-5, 44638-9, 60260-3 and 67320-6). The photomicrographs correspond to the first inoculation, followed by the 1st and 2nd passages. Sample 60260-3 showed atypical development of chlamydial growth from passage 1. Immunostaining was performed on the fixed infected cells at 48 h pi (20×). The first inoculation and passages were performed using the standard or the alternative growth protocol, as indicated in the figures. * indicates the presence of a fungus in the cell culture.

**Figure 2 microorganisms-13-00288-f002:**
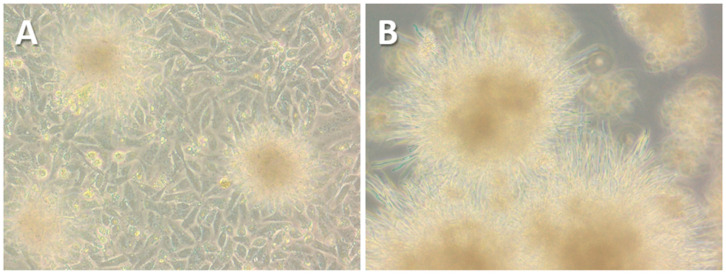
Photomicrographs (20×) showing the fungal-like structures contaminating the BGM cells infected with the sample 60260-3 at 24 h pi (**A**). The fungus was then cultured on Sabouraud’s dextrose agar plates supplemented with 0.5 g/L chloramphenicol (**B**).

**Table 1 microorganisms-13-00288-t001:** Summary of *C. gallinacea*-positive field samples (chicken cloacal swabs [3,13] selected for in vitro culture, using either the standard or the alternative protocols). PCR was performed using previously described PCR systems (^1^ [14] and ^2^ [15] (except for samples marked with *), which followed [16], before the improvement described in [15]).

Origin	Date of Collection	Specimen Identification	Cq *Chlamydiaceae* Real-Time PCR ^1^	Cq *C. gallinacea* Real-Time PCR ^2^	Culture Protocol
Mexico	May, 2018	18-2470/Ameca-1 (A1)	21.6	21.6	standard
Mexico	May, 2018	18-2470/Ameca-8 (A8)	20.8	21.4	standard
Mexico	February, 2018	18-2470/Pue6-6	22.4	23.6	standard
Mexico	February, 2018	18-2470/Pue6-7	25.1	26.0	standard
Italy	May, 2019	20-2327/41645-5	29.6	19.3 *	alternative
Italy	May, 2019	20-2327/44638-9	29.6	18.2 *	alternative
Italy	July, 2019	20-2327/60260-3	28.0	27.6	alternative
Italy	August, 2019	20-2327/67320-6	21.1	20.9	alternative

## Data Availability

All data are available upon request.

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
