# Peer review of "Trichosporon asahii: A Potential Growth Promoter for C. gallinacea? Implications for Chlamydial Infections and Cell Culture"

_microorganisms, 2025, doi:10.3390/microorganisms13020288_

Round 1
Reviewer 1 Report
Comments and Suggestions for Authors
Trichosporon asahii: a potential growth promoter for C. gallinacea? Implications for chlamydial infections and cell culture
The aim of this study was to test different culture conditions for the isolation of C. gallinacea from field samples, with the visualization of well-formed inclusions using immunofluorescence.
Several cell culture infection protocols were tested in this study to improve its growth, but none were successful for field samples. The only exception was an accidental co-culture with Trichosporon asahii, another microorganism naturally present in the positive sample. This suggests that current in vitro culture methods may not be suitable for this species and that other organisms in the host may influence the in vivo growth of C. gallinacea.
The paper is exciting, however, there are a few writing and formatting issues.
The methods are well explained.
Be consistent: Italic for all bacteria names, in vitro
(7.3 x 107 IFU/mL) and elsewhere
Author Response
Reviewer 1
The aim of this study was to test different culture conditions for the isolation of C. gallinacea from field samples, with the visualization of well-formed inclusions using immunofluorescence. Several cell culture infection protocols were tested in this study to improve its growth, but none were successful for field samples. The only exception was an accidental co-culture with Trichosporon asahii, another microorganism naturally present in the positive sample. This suggests that current in vitro culture methods may not be suitable for this species and that other organisms in the host may influence the in vivo growth of C. gallinacea. The paper is exciting, however, there are a few writing and formatting issues. The methods are well explained. Be consistent: Italic for all bacteria names, in vitro, (7.3 x 107 IFU/mL) and elsewhere
Reply: The modifications (italicization of bacterial names, in vitro terms, and using superscript for titles) have been made as requested throughout the manuscript.
Reviewer 2 Report
Comments and Suggestions for Authors
C. gallinacea growth did not lead to improved cultivation methods for better inclusion formation and passage propagation. However, despite laboratory constraints that prevented evaluation of the alternative protocol with the reference C. gallinacea strain,
the text above must be highlighted as limitation of this study.
The number of tests should be increased. Replication is very low. The conclusions cannot justify a publication in a scientific journal.
New and recent references may be added. Minor points :Authors must write C. gallinacea in italics, throughout the text.
Author Response
1. C. gallinacea growth did not lead to improved cultivation methods for better inclusion formation and passage propagation. However, despite laboratory constraints that prevented evaluation of the alternative protocol with the reference C. gallinacea strain, the text above must be highlighted as limitation of this study.
Reply: The sentence has been completed to take account of this comment (new version, Lines 221-223). However, despite laboratory constraints that prevented evaluation of the alternative protocol with the reference C. gallinacea strain – a limitation of this study - successful propagation was achieved in a field sample.
2. The number of tests should be increased. Replication is very low. The conclusions cannot justify a publication in a scientific journal.
Reply: We fully acknowledge that the comparative tests of different culture conditions were carried out on a limited number of trials. However, the propagation of C. gallinacea under current cell culture conditions (using either the standard or alternative protocol) remains suboptimal, which limits the number of trials that can be conducted due to limited inoculum availability. Nevertheless, the main focus of this article is the serendipitous discovery of a potentially important cofactor for C. gallinacea cultivation and the potential implications of this observation in the context of in vivo infection. For this reason, the methodological section has not been elaborated in detail and has instead been provided as an appendix. The discovery includes the identification of yeast that may play a critical role. Ongoing studies aim to confirm this hypothesis and the results will be presented in a more detailed second article. We believe that this finding is worth sharing with the scientific community.
3. New and recent references may be added.
Reply: New and recent references have been added.
4.Minor points: Authors must write C. gallinacea in italics, throughout the text.
Reply: This has been corrected throughout the manuscript.
Reviewer 3 Report
Comments and Suggestions for Authors
The paper effectively highlights the challenges in cultivating Chlamydia gallinacea and provides innovative insights through its discovery of Trichosporon asahii co-culture as a potential facilitator for chlamydial growth. By systematically testing various cell culture conditions, the study underscores the role of microbial interactions in enhancing in vitro growth, offering a new perspective on cultivating challenging bacterial species. The integration of experimental and observational data, particularly the detailed analysis of fungal co-culture effects, strengthens the argument for exploring co-infections in both laboratory and natural environments. This approach opens avenues for further research into the complex dynamics of pathogen-host-microbiome interactions, with implications for both public health and veterinary sciences.
Clarify the methodology for experimental protocols to ensure reproducibility, including precise descriptions of modifications to standard protocols.
Provide quantitative data on inclusion formation and growth comparisons across different experimental conditions.
Include statistical analysis to validate claims about the effect of Trichosporon asahii on C. gallinacea growth.
Elaborate on potential mechanisms by which T. asahii influences chlamydial growth with more experimental or literature-supported evidence.
Add a detailed discussion on the implications of fungal co-culture for in vivo pathogenicity and transmission risks.
Address any inconsistencies in results between different laboratories, providing potential explanations.
Expand the conclusion to include specific recommendations for future research and potential applications of findings.
Photomicrographs, micrographs, microphotographs: Use the same term, and provide a length bar for each of them.
Author Response
1.The paper effectively highlights the challenges in cultivating Chlamydia gallinacea and provides innovative insights through its discovery of Trichosporon asahii co-culture as a potential facilitator for chlamydial growth. By systematically testing various cell culture conditions, the study underscores the role of microbial interactions in enhancing in vitro growth, offering a new perspective on cultivating challenging bacterial species. The integration of experimental and observational data, particularly the detailed analysis of fungal co-culture effects, strengthens the argument for exploring co-infections in both laboratory and natural environments. This approach opens avenues for further research into the complex dynamics of pathogen-host-microbiome interactions, with implications for both public health and veterinary sciences. Clarify the methodology for experimental protocols to ensure reproducibility, including precise descriptions of modifications to standard protocols.
Reply: The differences between the two protocols are mentioned in the text (new version, lines 142-146) and summarized in Table S2. - "As a final approach, we tested an enriched culture medium inspired by Donati and colleagues [4] for direct application to field samples. After inoculation and centrifugation, the cell culture medium was replaced with EMEM supplemented with glucose, L-glutamine, cycloheximide and 20% of foetal calf serum (Table S2). This alternative protocol requires CO2 incubation after medium replacement."
2.Provide quantitative data on inclusion formation and growth comparisons across different experimental conditions.
Reply: As stated in the aim of our study, the aim was to visualise well-formed inclusions, not to provide quantitative data, as immunofluorescence is the first test performed to observe inclusions and determine if bacteria can be cultured. One limitation of C. gallinacea is that it does not form well-structured inclusions in some laboratories. This point has been clarified in the text, as it was previously not well presented and could lead to misunderstandings, especially for those unfamiliar with C. gallinacea (as few laboratories have successfully cultured this bacterium)(new version, lines 56-60). -
"However, in studies carried out in various laboratories to characterise new field isolates, it has been observed that, in some laboratories, their infectivity tends to decrease progressively with each subsequent passage in chicken embryos or BGM cells, with immunofluorescence analysis revealing small, diffuse inclusions in the form of a starry sky, ultimately leading to the loss of the strains (unpublished data)."
3.Include statistical analysis to validate claims about the effect of Trichosporon asahii on C. gallinacea growth.
Reply: This article describes the experimental context that led to the identification of the presence of yeast while well-formed inclusions of C. gallinacea were observed for a specific field sample. Further studies are underway to document this interaction in more detail and will form the basis of a future publication due to the effort required to purify this yeast strain and study their interaction in the context of an obligate intracellular bacterium.
4.Elaborate on potential mechanisms by which T. asahii influences chlamydial growth with more experimental or literature-supported evidence.
Reply: This has been evocated, further studies are needed to explained potential mechanisms.
New version, lines 185-190
A recent study has shown that the biofilm formed by Candida albicans may favor the survival of C. trachomatis within it, thereby maintaining its infectious properties [24]. This bio-film provides a protective environment that reduces chlamydial susceptibility to antibiotics and promotes evasion of the host immune system. Several Trichosporon species, including Trichosporon asahii, are capable of producing highly resistant biofilms [25].
New version, lines 208-218
This raises the possibility that small variations in sensitive reagents, such as fetal calf serum (which may differ in protein, vitamin, or mineral content), could be critical for C. gallinacea growth. These essential nutrients could be provided in vivo by Trichosporon, a yeast naturally present in poultry, in the context of co-culture as identified in our study. While yeast extracts are rarely used to culture of Chlamydia, they are commonly used for mycoplasmas, which also have small genomes and limited biosynthetic capabilities. Growth factors provided by yeast extracts or fetal calf sera [30] are not standardized and are likely to have varying nutrient content between batches, potentially leading to inconsistent results. Transcriptomic studies could help to identify the necessary supplements to be added, as recently described for Mycoplasma ovipneumoniae [31].
5. Add a detailed discussion on the implications of fungal co-culture for in vivo pathogenicity and transmission risks.
Reply: The in vivo fungal co-culture may provide essential nutrients or create an enabling environment for C. gallinacea, (an obligate intracellular bacterium with a very small genome) to promote its infectivity. This has been addressed in the text.
New version, lines 185-190
A recent study has shown that the biofilm formed by Candida albicans may favor the survival of C. trachomatis within it, thereby maintaining its infectious properties [24]. This bio-film provides a protective environment that reduces chlamydial susceptibility to antibiotics and promotes evasion of the host immune system. Several Trichosporon species, including Trichosporon asahii, are capable of producing highly resistant biofilms [25].
New version, lines 210-212
These essential nutrients could be provided in vivo by Tricho.
6.Address any inconsistencies in results between different laboratories, providing potential explanations.
Reply: This point is addressed in the text (new version, lines 204-210). Fetal calf serum is identified as a critical factor for Chlamydia cultivation, and differences between batches have already been observed. The fetal calf serum used at Anses was shown to be suitable for the cultivation of standard Chlamydia, but not for C. gallinacea. This distinction (between species) had not been previously recognized, but it is now clearly stated in the paper.
"This could also explain why inclusions are observed during the initial cultivation of a field sample but disappear during subsequent passages. The C. gallinacea reference strain has been successfully propagated on a BGM cell line for up to 10 passages at the FLI in Germany, in contrast to the results obtained at Anses in France using an identical protocol (standard protocol). This raises the possibility that small variations in sensitive reagents, such as fetal calf serum (which may differ in protein, vitamin, or mineral content), could be critical for C. gallinacea growth."
7.Expand the conclusion to include specific recommendations for future research and potential applications of findings.
Reply: The conclusion has been expanded (new version, lines 233-241).
"It is now important to investigate how the yeast-depleted strain behaves in vitro and to assess the effect of this yeast on other C. gallinacea strains. It would also be interesting to determine whether this effect is specific to Trichosporon or whether other yeasts could have a similar effect. Several Chlamydia species have recently been described in birds, suggesting host specificity (e.g. C. gallinacea in chickens, Candidatus C. ibidis in flamingos, C. buteonis in birds of prey, etc.). Given the small genomes of chlamydiae, it is possible that these bacteria have evolved according to their environment, in particular the gut microbiota in which they persist, and possibly according to the metabolites available in that environment."
8.Photomicrographs, micrographs, microphotographs: Use the same term, and provide a length bar for each of them.
Reply: Photomicrographs were used throughout the text. While the scale bar was not provided, the magnification (x20) is mentioned.
Round 2
Reviewer 2 Report
Comments and Suggestions for Authors
Reduce the similarity rate of the text by less than 15%.
Abstract
The abstract highlights the difficulty in cultivating C. gallinacea using existing in vitro protocols. Τhe lack of details on the tested protocols makes it difficult to evaluate why they failed.
The abstract raises the important question of how microbial interactions influence the pathogenicity of chlamydial species. However, it does not clarify whether C. gallinacea is pathogenic or commensal in its host.
Short communiacation
The phrase "with a unique life cycle that alternates between extracellular, infectious elementary bodies and intracellular, non-infectious reticulate bodies" is scientifically correct but could be made more concise by explicitly linking the forms to their roles (e.g., transmission and replication).
The sentence "linking the presence of chlamydia to the infectious status, of the farm and suggesting a possible bird-to-human transmission" is unclear. It should be rewritten.
The statement that C. gallinacea has "recently been detected in poultry farmers" is significant but requires more specifics.
The study aims to test different culture conditions for C. gallinacea isolation is clearly stated. However, no details are provided about the specific conditions tested or their rationale.
Line 69: CO₂ (not CO2, as the 2 should be in subscript form)
Were control experiments conducted with other fungal isolates or field samples to determine if this atypical growth was indeed due to T. asahii or another variable?
While the protocol for immunofluorescence labelling is described, its advantages (e.g., sensitivity, ability to visualize inclusions) are not discussed. Briefly explain why this method was chosen over alternatives.
Conclusion
Line 231: The phrase "successful propagation was achieved in a field sample" has to be explained.
Line 247: "Given the small genomes of chlamydiae, it is possible that these bacteria have evolved according to their environment, particularly the gut microbiota, and the metabolites available in that environment." This sentence is unclear.
Expand on future research directions
Comments on the Quality of English LanguageEnglish language has to be improved.
Author Response
Please see the attached reply.
